# Screening of Differentially Expressed Genes and miRNAs in Hypothalamus and Pituitary Gland of Sheep under Different Photoperiods

**DOI:** 10.3390/genes13061091

**Published:** 2022-06-19

**Authors:** Qingqing Liu, Ran Di, Chunhuan Ren, Xiaoyun He, Xiangyu Wang, Qing Xia, Mingxing Chu, Zijun Zhang

**Affiliations:** 1College of Animal Science and Technology, Anhui Agricultural University, Hefei 230036, China; qingqingliu04@163.com (Q.L.); renchunhuan@126.com (C.R.); 2Key Laboratory of Animal Genetics, Breeding and Reproduction of Ministry of Agriculture and Rural Affairs, Institute of Animal Science, Chinese Academy of Agricultural Sciences, Beijing 100193, China; dirangirl@163.com (R.D.); hexiaoyun@caas.cn (X.H.); wangxiangyu@caas.cn (X.W.); xiaqingxx@126.com (Q.X.)

**Keywords:** sheep, photoperiod, OVX + E_2_ model, whole transcript sequencings, miRNAs

## Abstract

The reproduction of sheep is affected by many factors such as light, nutrition and genetics. The Hypothalamic-pituitary-gonadal (HPG) axis is an important pathway for sheep reproduction, and changes in HPG axis-related gene expression can affect sheep reproduction. In this study, a model of bilateral ovarian removal and estrogen supplementation (OVX + E_2_) was applied to screen differentially expressed genes and miRNAs under different photoperiods using whole transcriptome sequencing and reveal the regulatory effects of the photoperiod on the upstream tissues of the HPG axis in sheep. Whole transcriptome sequencing was performed in ewe hypothalamus (HYP) and distal pituitary (PD) tissues under short photoperiod 21st day (SP21) and long photoperiod 21st day (LP21). Compared to the short photoperiod, a total of 1813 differential genes (up-regulation 966 and down-regulation 847) and 145 differential miRNAs (up-regulation 73 and down-regulation 72) were identified in the hypothalamus of long photoperiod group. Similarly, 2492 differential genes (up-regulation 1829 and down-regulation 663) and 59 differential miRNAs (up-regulation 49 and down-regulation 10) were identified in the pituitary of long photoperiod group. Subsequently, GO and KEGG enrichment analysis revealed that the differential genes and target genes of differential miRNA were enriched in GnRH, Wnt, ErbB and circadian rhythm pathways associated with reproduction. Combined with sequence complementation and gene expression correlation analysis, several miRNA-mRNA target combinations (e.g., LHB regulated by novel-414) were obtained. Taken together, these results will help to understand the regulatory effect of the photoperiod on the upstream tissues of HPG in sheep.

## 1. Introduction

Animals can adapt themselves to their survival environment by regulating themselves, for example, females will choose to breed in specific seasons to ensure that they give birth in the most appropriate season, which will facilitate lactation and early pup development, and a stable physiological mechanism has evolved over the years [1]. Light perception is one of the important mechanisms that trigger the response of organisms and many physiological processes in mammals are influenced by the photoperiod [2]; a long photoperiod induces circadian transcription factor *BMAL2* in pars tuberalis (PT) and triggers summer biological activity through the *EYA3*/*TSH* pathway. In contrast, the continuous prolongation of a short photoperiod leads to melatonin secretion inducing circadian repressors, including *DEC1*, which inhibit *BMAL2* and *EYA3*/*TSH* pathways, into winter biological activity, and circadian rhythms are usually involved in some way in photoperiod-induced physiological activity [3]. Seasonal changes in light have profound effects on the behavior and physiological functions of many animals, and changes in the light-dark cycle can also induce anxiety and depressive behavior in adult rodents [4,5]. The development of the visual system is affected by light, and mice kept in the dark show alterations in a retinal synaptic organization and visual function [6]. In short photoperiods, birds can meet increased energy demands in winter by increasing their body weight, digestible energy intake and digestive tract size [7], and photoperiods also induce migratory behavior in birds [8]. Photoperiodic changes also influence the neuroendocrine action in fish [9] and horse deer [10]. Circadian clock genes are involved in seasonal activity, and clock genes are expressed in PT and ventricular canal cells within the hypothalamus [11], for example, the ability to regulate seasonal changes in reproductive dormancy is disrupted in drosophila, which is deficient in the *EYA* gene [12]. Many studies on photoperiodic effects on reproduction have been carried out on quail and hamsters [13,14,15].

Reproduction in sheep is a complex biological process influenced by many factors such as light, nutrition, genetics, reproductive hormones, and reproductive techniques. The reproduction of mammals was controlled by the hypothalamic-pituitary-gonadal (HPG) axis regulation network of the reproductive axis. The supraoptic nucleus receives light signals to the pituitary gland, where light signaling factors and cell morphology in the pituitary nodules are transformed, leading to morphological changes in hypothalamic GnRH neurons and ultimately to changes in the levels of gonadotropins secreted by the distal pituitary gland, which ultimately induce estrus and estrus in sheep [16,17,18,19]. Early recognition of the role of the hypothalamus as an endocrine organ and as a site of integration of autonomic and behavioral responses dates back centuries, and although the hypothalamus represents only 2% of the total volume of the brain, it is a key regulator of pituitary function and homeostatic balance, - it therefore has many functions, including food and fluid intake, lactation, thermoregulation, circadian rhythms and sleep-wake cycles [20]. The pituitary gland is targeted and regulated by the hypothalamic fine cell neuroendocrine system through a hormonal feedback mechanism. The anterior pituitary is a key organ involved in the control of a variety of physiological functions, including growth, reproduction, and metabolic functions. To perform its functions, the pituitary receives and processes signals from the central and peripheral sources and transmits them appropriately to several key endocrine and non-endocrine organs [21]. These cells produce prolactin (PRL), growth hormone (GH), corticotropin (ACTH), gonadotropin, thyroid-stimulating hormone, and follicle-stimulating hormone (FSH)/luteinizing hormone (LH), which act on the organism. Therefore, it is important to investigate the genetic changes in the hypothalamus and pituitary gland related to estrus and circadian rhythms for reproduction in sheep.

With the development and application of whole transcriptomics sequencing technology, many miRNAs have been identified in various tissues and organs of sheep. In recent years, a large number of miRNAs are involved in reproduction in both sheep [22,23] and other species [24,25,26]. The lncRNA of different light-induced reproduction in sheep have been mined in our laboratory in the previous stage using the classical model of the OVX + E_2_ model [27,28]. The study used this model to explore the regulatory role of the upstream tissues of the HPG axis in sheep and to screen the hypothalamic and pituitary differential genes and miRNAs in sheep with different photoperiods, which will lay the theoretical foundation for revealing the molecular mechanism of photoperiodic regulation of estrus in sheep.

## 2. Materials and Methods

### 2.1. Animals and Sample Collection

A group of 21 Small Tail Han ewes (3 y, clinically normal, and non-pregnant) were selected from Heilongjiang Province, China. All ewes were fed ad libitum and had free access to water. After removal of the ovaries (OVX) from the experimental Small Tail Han ewes with good estrous traits as described above, estrogen silicone tubes were implanted subcutaneously, and the sheep were penned into artificially controlled light sheds after the hormone concentration was stabilized at 7.23 ± 2.50 pg/mL. OVX + E_2_ sheep and light control rooms were constructed as previously described [27]. After 21 days of short photoperiod (8/16 h light-dark), six were euthanized (intravenous pentobarbital 100 mg/kg). After slaughtering, hypothalamic and distal pituitary tissues were taken and were labeled as SP21−HYP and SP21−PD, respectively. Six other Small-tailed Sheep were transferred to a long photoperiod (16/8 h light-dark) sheep barn and after 21 days of rearing, tissues were slaughtered and collected and labeled LP21−HYP and LP21−PD. All collected tissue was snap-frozen at liquid nitrogen, and then brought back to the laboratory and stored in a −80 °C refrigerator for long-term storage.

### 2.2. RNA Extraction, Library Construction, Sequencing and Raw Data Processing 

Total RNA from each sample was isolated using the TRIzol reagent (Invitrogen, Carlsbad, CA, USA). then the Nanodrop 2000 instrument was used to detect the concentration and purity of the extracted RNA, and the integrity of RNA was detected by agarose gel electrophoresis. After the samples were tested and passed, cDNA and sRNA library construction were performed according to the previous method [29]. To finally obtain valid data, we filtered the raw sequencing data to obtain clean reads. Clean reads were aligned to the sheep reference genome using bowtie (Oar_rambouillet_v1.0).

### 2.3. Differential Expression Analysis of mRNAs

Quantitative analysis was performed using StringTie software to obtain the number of Reads per sample compared to each transcript, which was converted to FPKM (Fragments Per Kilobase per Million bases). Quantitative statistical analysis of transcript expression was performed to filter out transcripts with significantly different expression levels in the samples. We used DESeq2 to analyze the significance of transcript expression differences. The differentially expressed transcripts were screened from two levels of fold change (FC) and corrected significance level (padj/*q*-value). In addition, |log_2_FC| > 1 and *q-*value < 0.05 were considered DEGs.

### 2.4. Differential Expression Analysis of miRNAs and Target Gene Prediction

Identification of known miRNAs was performed using bowtie software by comparing sRNAs to miRNA sequences from the specified range of miRBase (v22) database, and prediction of new miRNAs was performed using mirEvo and miRDeep software and expression information was obtained. The expression of miRNAs represented by TPM was obtained by comparing to the reads of miRNA precursor and mature body regions. After calculating the expression, differential analysis was performed using DESeq2, and the screening threshold for differential miRNAs was set to |log_2_FC|> 1, *p* value < 0.05.

The previously obtained known miRNAs and new miRNAs were analyzed for miRNA target gene prediction. The miRanda and qTar software were selected for target gene prediction, and the intersection of the two software was taken as the final result.

### 2.5. Functional Annotation and Enrichment Analysis of Target Genes of DE miRNAs and DE mRNAs

The DEGs and the predicted target genes of DEMs were analyzed by Gene Ontology (GO) (referred to as GO, http://www.geneontology.org/, accessed on 6 January 2022), which includes biological process, molecular function and cellular component, and Kyoto Encyclopedia of Genes and Genomes (KEGG). GO items or KEGG pathways with a hypergeometric *p*-value < 0.05 were those that were significantly enriched.

### 2.6. Construction of Integral miRNA-mRNA Interaction Networks

To accurately identify the key association with reproductive DEMs and DEGs, based on miRNA function, the mRNAs that were negatively related to miRNAs were screened out, and the miRNA-mRNA interaction networks were built by using Cytoscape (v3.8.2, http://www.cytoscape.org/, accessed on 1 April 2022).

### 2.7. Quantitative PCR Validation

To verify the accuracy of sequencing data, 7mRNAs including hypothalamus (*DIO2*, *SIX1*, *RASD1*, *KCNH3*) and distal pituitary (*OXT*, *THUMPD1*, *LOC101114319*) were randomly selected. The first primers were synthesized by Bioengineering (Shanghai) Co., Ltd. For the qPCR analysis of mRNAs, reverse transcription was performed using the PrimeScriptTM RT reagent kit (TaKaRa, Dalian, China). Furthermore, qPCR with the SYBR Green qPCR Mix (TaKaRa, Dalian, China) was conducted with a RocheLight Cycler480^®^ Ⅱ system (Roche Applied Science, Mannheim, Germany) as follows: initial denaturation at 95 °C for 5 min, followed by 40 cycles of denaturation at 95 °C for 5 s, then annealing at 60 °C for 30 s. The sequences of qPCR primers are listed in Table 1.

Eight miRNAs were randomly selected, including hypothalamus (oar-miR-200a, novel_10, novel_101, novel_4, oar-miR-143) and distal pituitary (novel_142, oar-miR-181a, oar-miR-323b). For the qPCR analysis of miRNAs, reverse transcription was performed using miRcute Plus miRNAs First-Strand cDNA Kit (TIANGEN, Beijing, China). Then, qPCR was conducted miRcute Plus miRNA qPCR Kit (TIANGEN, Beijing, China) using a RocheLight Cycler480^®^ II system (Roche AppliedScience, Mannheim, Germany) in the following procedure: initial denaturation at 95 °C for 15 min, followed by 40 cycles of denaturation at 94 °C for 20 s, then annealing at 60 °C for 34 s. The primers specific to miRNAs were synthesized by Beijing Tianyi Huiyuan Biotechnology Co., Ltd. (Beijing, China). The primer design information is shown in Table 2.

In addition, R*PL19* (for mRNA) and U6 small nuclear RNA (for miRNA) were utilized as reference gene-miRNA to calculate the relative expression level with the method of 2^−ΔΔCt^.

## 3. Results

### 3.1. Summary of Sequencing Data for mRNA and miRNA

RNA-seq for mRNA generated about 673 million (hypothalamus) and 640 million (distal pituitary) clean reads after data filtering, and more than 94% of the clean reads were located in the genome. The GC content of clean reads was between 47.97 and 53.37%, and the clean reads quality scores of Q20 and Q30 were above 96.29% and 89.87%, respectively, indicating that the reliability and quality of the sequencing data were sufficient for further analysis (Appendix A). Regarding the expression level of mRNAs, details are provided in Appendix A, Figure 1A,C. In addition, the chromosomal distribution of mRNA on both tissues showed that genes were mostly present on chromosomes 1, 2 and 3, occupying 27.53% and 27.46%, respectively, (Appendix A, Figure 1B,D).

RNA-seq for miRNAs generated about 140 million (hypothalamus) and 132 million (distal pituitary) clean reads after data filtering, and in the range of 92.99–98.65% of the clean reads, which were located in the genome (Appendix A), with the length between 21–24nt (Appendix A, Figure 2B,D). The average Q20 content was 98% (hypothalamus) and 96% (distal pituitary) (Appendix A). In addition, a variety of non-coding RNAs (ncRNAs) were also identified, including transfer RNAs (tRNAs), snRNAs and miRNAs (Appendix A, Figure 2A,C). The known miRNAs accounted for only a part of all identified ncRNAs, 29.85% and 32.73%, respectively. However, most of them are newly identified miRNAs. Totally, 902 new miRNAs and 149 known miRNAs were included in the hypothalamus, 628 new miRNAs and 152 known miRNAs in the distal pituitary (Appendix A).

### 3.2. Identification of Differential Expressed Genes and miRNAs

A total of 1813 DEGs were identified from SP21−HYP *vs.* LP21−HYP comparison. Compared to the short photoperiods,966 genes were up-regulated, and 847 genes were down-regulated in the long photoperiods (Appendix A, Figure 3A). Similarly, a total of 2492 DEGs were identified from SP21−PD *vs.* LP21−PD comparison. Among these DEGs, 1829 genes were up-regulated, and 663 genes were down-regulated in the long photoperiods (Appendix A, Figure 3C). The 10 mRNAs with the most significant difference are shown in Table 2.

A total of 145 DEMs were detected among the 1051 identified miRNAs. Compared to the short photoperiods, 73 were up-regulated and 72 were down-regulated in the long photoperiods (Appendix A, Figure 3B). A total of 59 DEMs were detected among the 1051 identified miRNAs, of which 49 were up-regulated and 10 were down-regulated (Appendix A, Figure 3D). The 10 mRNAs with most significant difference was shown in Table 3, the 10 miRNAs with most significant difference was shown in Table 4, Sequence details of miRNAs are available at Appendix A.

### 3.3. Functional Enrichment Analysis of the DEGs 

GO and KEGG analyses were conducted for the differentially expressed mRNAs. In GO terms, the most enriched terms in SP21−HYP *vs.* LP21−HYP are actin cytoskeleton organization and protein phosphorylation (Appendix A, Figure 4A). KEGG analysis showed that MAPK signaling pathway, GnRH signaling pathway, and insulin signaling pathway were significantly enriched (Appendix A, Figure 4C).

In GO terms, the most enriched terms in SP21−PD *vs.* LP21−PD are modulation of chemical synaptic transmission and regulation of trans-synaptic signaling (Appendix A, Figure 4C). All of these pathways are related to the basic activities of life, such as circadian entrainment, oxytocin signaling pathway and GnRH signaling pathway, and are associated with reproduction (Appendix A, Figure 4D). 

### 3.4. miRNA-mRNA Interaction Network

To further analyze the relationship between miRNAs and mRNAs, we constructed a genomic network of interactions using potential target genes of DEMs with DEGs obtained by RNA-Seq. Crossing the identified potential target genes of DEMs with the DEGs obtained from RNA-Seq, there were 468 hypothalamic overlapping genes and 426 pituitary overlapping genes (Figure 5). Then, Cytoscape was used to construct a co-regulatory network of DE miRNA-mRNA pairs with multiple targeting relationships (Figure 6, Appendix A).

To further determine the role of genes and miRNAs in the regulation of the gonadal axis by light conversion. GO and KEGG analysis was undertaken for the above overlapping genes (i.e., simultaneous differential expression of miRNAs and target genes). The two most enriched entries of GO are the regulatory activity of GTPase regulator activity and nucleoside-triphosphatase regulator activity in SP21−HYP *vs.* LP21−HYP (Appendix A, Figure 7A). GO results found that positive regulation of glial cell production was significantly enriched in SP21−PD *vs.* LP21−PD (Appendix A, Figure 7B). The pathways of KEGG enrichment of target genes in DEMs and DEGs were similar in both groups (Appendix A, Figure 7C, Appendix A, Figure 7D).

### 3.5. Data Validation

In order to assess the accuracy of sequencing, mRNAs and miRNAs were selected randomly for qPCR validation. We measured the gene expression level and compared it with the RNA-seq data. The results demonstrated that RNA-seq data and qPCR data showed similar patterns (Figure 8), which indicate the reliability of the data generated from RNA-seq.

## 4. Discussion

In most animals, reproduction is regulated by the hypothalamic-pituitary-gonadal axis [30]. The hypothalamus and pituitary gland are important upstream tissues that regulate HPG in sheep and control a wide range of animal behaviors [31,32]. To better probe the effects of the photoperiod on hypothalamic and pituitary function, we applied the OVX + E_2_ sheep model, which is often used to study the response of seasonally breeding animals to photoperiodic and reproductive endocrine changes [33,34,35]. Therefore, this study used transcriptomic approaches to mine differential genes and miRNA expression patterns associated with reproduction in sheep with different photoperiods.

### 4.1. Identification and Analysis of mRNAs and miRNAs Data in SP21− HYP vs. LP21− HYP

A total of 145 DEMs were identified for SP21−HYP *vs.* LP21−HYP, including miR-200 [36]. Family members were identified multiple times in seasonal and non-seasonal estrus ovaries of sheep. Among them, miR-200a is a member of the miR-200 family, which is associated with ovarian development in goats [37]. One study showed that miR-494-3p significantly inhibited apoptosis of endometrial epithelial cells [38], and oar-miR-485-3p can regulate the developmental quality of oocytes [39]. From the above, we can speculate that DEMs target and regulate genes related to reproductive estrus, which may be responsible for the reproductive variation in sheep due to different photoperiodic changes.

A total of 1813 DEGs were identified in SP21−HYP *vs.* LP21−HYP, including *APAF1*, *CASP3* associated with apoptosis, and *DIO2*, *LHB*, and *TSHB* associated with estrous reproduction in sheep. Some studies have demonstrated that haplotype H2 and diploid H2/H4 of TSHB may be associated with year-round estrus [40]. GO was enriched for many terms associated with intercellular communication, and differential genes expressed may influence changes in reproductive traits in sheep by affecting the expression of intercellular communication-related receptor proteins. KEGG pathways include enriched ErbB signaling pathway; GnRH signaling pathway (*PTK2B* and *CAM2B*), gonadotropin secretion (LHB), and prolactin signaling pathway were also significantly enriched; these pathways have been studied and proven to be associated with reproductive activity [27,41].

### 4.2. Analysis of miRNA-mRNA Interaction Network in SP21−HPY vs. LP21−HPY

To better understand the function of miRNAs, the SP21−HPY and LP21−HPY interaction negative network was constructed. In the upregulated network, a key oar-miR-200a targeting *CACNA2D1* was identified, and it has been shown that miR-200a-3p plays an important role in chicken [42] and goat [37] reproductive regulation, and some other miRNAs target genes that regulate cell proliferation apoptosis-related biological processes [43].

Among the down-regulated networks, the important DEGs and DEMs were selected to construct the reciprocal network, in which *DENND1A* was the target gene of novel_550, and it was shown that *DENND1A* gene was expressed higher in the low yield group than in the high yield group in ovarian and showed an opposite expression trend with chi-miR-324-3p. CCK-8 assay showed that chi-miR-324-3p overexpression significantly inhibited the proliferation of GCs and knockdown of chi-miR-324-3p promoted GC proliferation [44], thus affecting ovulation and reproduction in ewes, while novel_550 inhibited *DENND1A* gene expression in this study. We speculate that novel_550 and chi-miR-324-3p act similarly. It has also been reported that *DENND1A* is a susceptibility gene for polycystic ovaries, which is essential for ovarian and embryonic development [45,46,47] and can affect the reproductive process. Another target gene of novel_550, *EML6* was also shown to be associated with reproduction. The production of high-quality eggs in mouse reproduction requires normal segregation during oocyte meiosis, and *EML6* is highly expressed in oocytes and is responsible for the accurate segregation of homologous chromosomes [48]. GH is a target gene of novel_984; GH plays an inductive role in ovulatory reproduction [49], and is elevated during growth hormone secretion during major transitions in reproductive status such as puberty and pregnancy [50]. It is an important regulator of female reproduction, and is involved in gonadal steroidogenesis, gametogenesis and ovulation [51], and alterations to the GH axis can have a reproduction. In growth regulation, it is mainly used with growth hormone (GH) to promote cellular metabolism, promote growth and regulate reproduction [52]. Novel_414 targets LHB, luteinizing hormone (LH), also known as luteinizing hormone, which is released by gonadotropins and promotes blood flow and secretion of progesterone in the ovary, affecting ovulation [53]. *SIX1* is a common target gene of novel_618 and novel_424, and long light signaling molecules are mainly mediated by EYA3 and TSHβ. *EYA3* can form a complex with thyrotropic embryonic factor (TEF) and *SIX1*, which synergistically promotes TSHβ transcription upon binding to the TSHβ promoter D element [54]. In summary, in the down-regulated network, the expression of the above DEGs in this study was significantly higher during the shift from short to long photoperiods, and we hypothesize that DEMs such as chi-miR-324-3p, novel_550, and novel_414 lead to alterations in reproductive traits in sheep by negatively regulating and reproduction-related genes.

### 4.3. Identification and Analysis of mRNAs and miRNAs Data in SP21−PD vs. LP21−PD

A total of 2942 DEGs were identified for SP21−PD *vs.* LP21−PD, including *CREB3L1* and *OXT*. *CREB3L1* acts as a downstream effector of TSH to regulate the expression of transport proteins and increase the ability of secretory amplification [55]. Under short photoperiods, the pituitary nodules secrete the thyrotropic hormone, which stimulates T3 in the hypothalamus, which in turn affects the secretion of gonadotropin-releasing hormone and gonadotropin. Studies have shown that *OXT* and its receptor (*OXTR*) play a central role in reproduction and metabolism, especially in females, by functioning in an estrogen-dependent manner in rats [56] and are associated with human social and emotional behavior, as well as physical and mental health and disease [57,58,59]. In addition, we performed functional enrichment analysis of target genes of DEGs and DEMs in sheep, respectively. The results of KEGG pathway enrichment in both showed many common pathways in LP and SP comparison, among which the most interesting reproductive pathways included GnRH signaling pathway, Wnt signaling pathway, ErbB signaling pathway, and MAPK signaling pathway. Circadian rhythm-related pathways included circadian and TNF signaling pathways, and these results are consistent with the Kazakh sheep [60], suggesting that reproduction in Small-tailed Sheep is also influenced by the photoperiod. In KEGG analysis, enriched for pathways related to calcium regulation, light influences vitamin D synthesis and thus changes in calcium processing expression. It was shown that the rapid, light-dependent changes in melatonin levels in the pineal organ are apparently regulated by novel calcium signaling pathways [61,62].

We found that among the identified miRNAs, the most highly expressed ones, oar-miR-148a and oar-miR-379-5p, oar-miR-495-3p, oar-miR-143, oar-miR-106b and oar-miR-218a, have been shown to play important roles in regulating GnRH release [63]. In addition, a variety of miRNAs are expressed in the sheep pituitary and at different levels under different light conditions. However, some miRNAs are species-specific, which is also likely to lead to different sensitivity to photoperiods in different animals and different PD secretion of hormones leading to differences in reproductive cyclicity. A total of 6 of the 59 DEMs are known, of which oar-let-7c, oar-let-7a, and oar-let-7b are of the same miRNA family, in animal genomes. The high conservation of let-7 in different animal species suggests that they may play important (and possibly similar) roles in biological processes in various organisms [64,65]. The effect of miR-150 on GC cell apoptosis by targeting the *STAR*, thereby affecting ovarian function and subsequently hormone secretion related to estrus [66].

### 4.4. Analysis of miRNA-mRNA Interaction Network in SP−PD vs. LP−PD

In the up-regulated network, containing 45 mRNAs and 14 miRNAs negative interactors, of which *ZNF787* and *CAMK2* are both target genes of novel_146, *CAMK2b* is one of the most prominent isoforms of *CAMK2* [67], and calcium/calmodulin-dependent protein kinase II (*CAMK2*) is a key player in synaptic plasticity and memory formation. It has been shown that *CAMK2b* is associated with apoptosis and is involved in the protection of neurons from homocysteine-induced apoptosis through the HIF-1α signaling pathway [68] and that *ZNF787* is a neuronal inhibitory molecule [69].

The down-regulated network contained 169 mRNAs and 6 miRNAs. *Bok* was targeted by novel_142, a pro-apoptotic Bcl-2 protein, and the loss of Bok significantly reduced susceptibility to apoptosis and had an important role in ovarian and follicular maturation and apoptosis in humans [70] and mice [71]. Therefore, we hypothesize that miRNAs are affecting estrus-related hormone activity by inhibiting apoptosis and thus ultimately estrus. Interestingly, novel_414, novel_427 and novel_587 collectively target *Lhx1*, and *Lhx1* deficiency causes severe loss of circadian rhythm and sleeps’ light control in mice [72], and in the present study, *Lhx1* expression was elevated during long photoperiods, in which we hypothesize that light sensitivity in sheep is reduced through miRNA regulation of *Lhx1*.

The two pathways that regulate seasonal estrus in sheep, are the hypothalamic-pituitary-gonadal axis and *KISS1*/*GPR54*. In birds [73] and sheep [74] TSHβ, a molecular signature of long photoperiods, increased in expression in the shift from short to long photoperiods. Under long photoperiods t, high levels of *TSH*β bind to *TSHR* and then drive the conversion of *DIO3* to *DIO2* through the cAMP signaling pathway, which regulates GnRH neurons, in addition to higher levels of *KISS1* expression in the sheep hypothalamus than in the non-breeding season [75]. In the present study, we did a collated comparison of differential genes in the comparison groups of hypothalamus and distal pituitary, and found consistent trends in *KISS1* and *DIO3* (Appendix A). In this study, during the short to long photoperiod shift, TSHR expression increased, *KISS1* gene expression decreased, *DIO2* significantly increased, and *DIO3* was decreasing, consistent with the above studies. *LHB* expression decreased and PRL increased, consistent with the trend of change during the shift from short to long photoperiods in seasonal estrus [60,75], but FSHR expression was slightly decreased, and these changes may also be the reason for the Small Tail Han sheep.

## 5. Conclusions

The present study provides, for the first time, the complete miRNA-mRNA network of sheep under photoperiodic regulation from the hypothalamus and pars tuberalis. We identified many DEGs (e.g., *LHB*, *TSHB*) and miRNA-mRNA pairs (e.g., *LHB* is a targeting gene of novel_414) from the RNA-seq data obtained from hypothalamus. We identifiedmany DEGs (e.g., *OXT*, *GH* and *Lhx1*) and miRNA-mRNA pairs (e.g., *Lhx1* is a targeting gene of novel_414, novel_427, and novel_587) from the RNA-seq data obtained from pars tuberalis. These data provide significant data for further exploring the reproduction of sheep.

## Figures and Tables

**Figure 1 genes-13-01091-f001:**
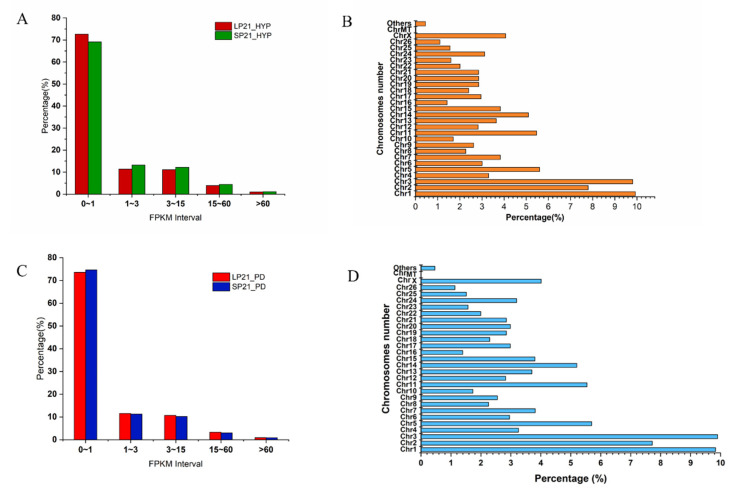
FPKM (**A**) and Chromosome (**B**) distribution of identified g expressed genes in SP21−HYP *vs.* LP21−HYP; FPKM (**C**) and Chromosome (**D**) distribution of identified expressed genes in SP21−PD *vs.* LP21−PD.

**Figure 2 genes-13-01091-f002:**
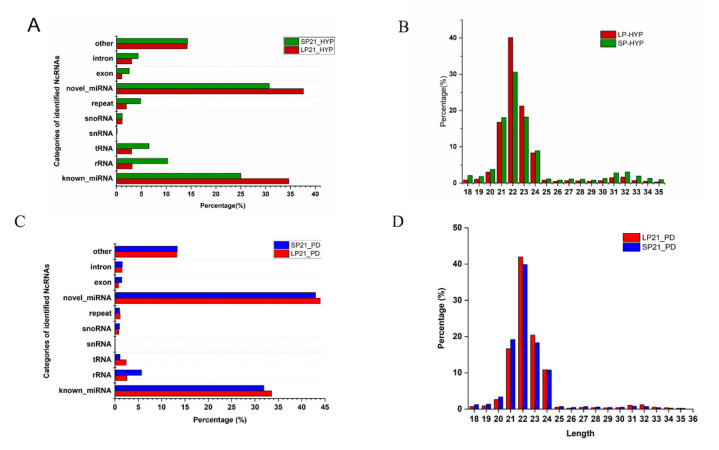
Types of non-coding rRNAs (**A**)and length distribution of sRNAs (**B**) from SP21−HYP *vs.* LP21−HYP, (**C**,**D**) from SP21−PD *vs.* LP21−PD.

**Figure 3 genes-13-01091-f003:**
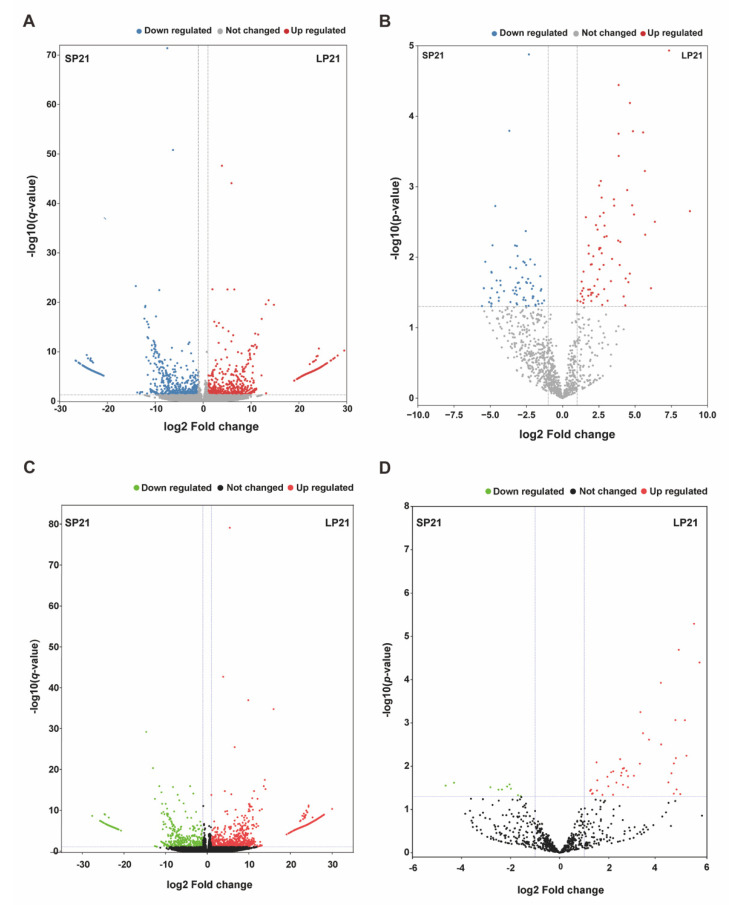
DEGs and DEMs analyses. Volcano plot of identified genes (**A**) and miRNAs (**B**) in SP21−HYP *vs.* LP21−HYP, where red and blue represent up or down-regulation, respectively. Volcano plot of identified genes (**C**) and miRNAs (**D**) in SP21−PD *vs.* LP21−PD, where red and green represent up or down-regulation, respectively.

**Figure 4 genes-13-01091-f004:**
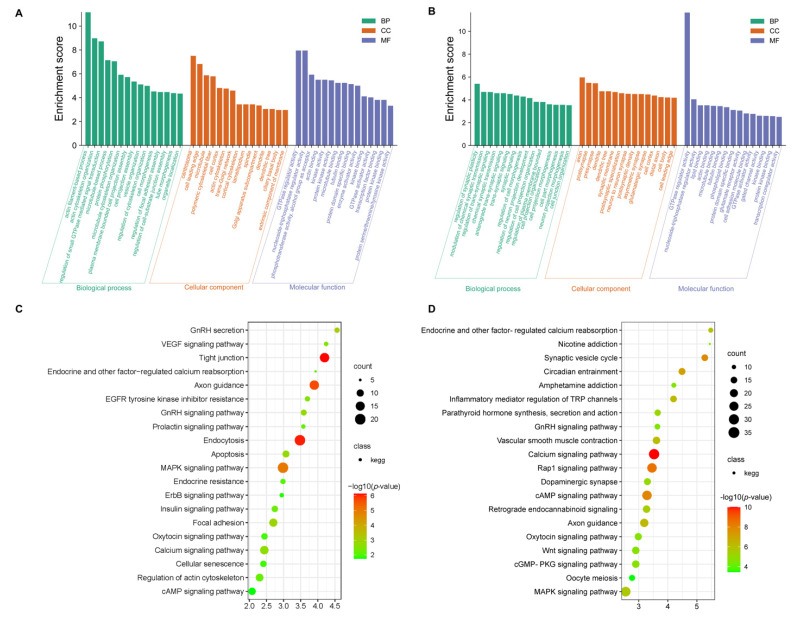
TOP enriched gene ontology (GO) study of differential expressed genes between short-photoperiod and long-photoperiod in the hypothalamus and distal pituitary of sheep. (**A**) GO enrichment terms for DEGs in SP21−HYP *vs.* LP21−HYP. (**B**) GO enrichment terms for DEGs in SP21−PD *vs.* LP21−PD. KEGG pathway analysis for DEGs between short photoperiod and long photoperiod in hypothalamus and PD of Small Tail Han sheep. (**C**) 20 enrichment pathways in SP21−HYP *vs.* LP21−HYP. (**D**) 20 enrichment pathways in SP21−PD *vs.* LP21−PD. Rich factors are defined as the amount of differentially expressed genes enriched in the pathway/amount of all genes in the background gene set.

**Figure 5 genes-13-01091-f005:**
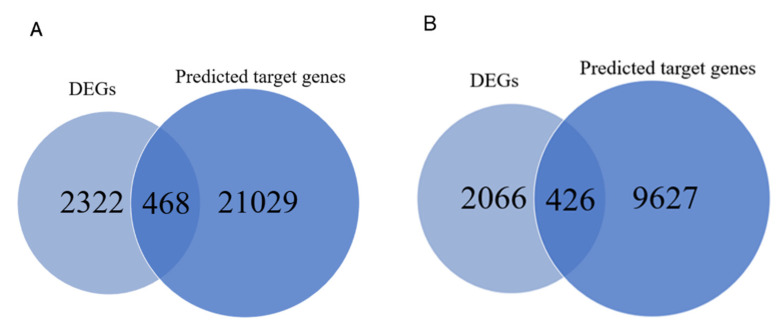
Overview of mRNA-miRNA networks. (**A**) Overlapping genes in SP21−HYP *vs.* LP21−HYP between DEGs and predicted target genes by miRNAs. (**B**) Overlapping genes in SP21−PD *vs.* LP21−PD between DEGs and predicted target genes by miRNAs.

**Figure 6 genes-13-01091-f006:**
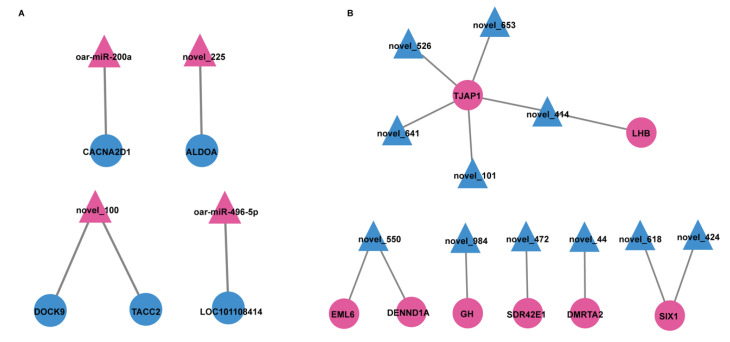
Regulatory networks of DE miRNA-mRNA in hypothalamus and distal pituitary of sheep. (**A**) The network contained 4 up-regulated miRNAs and 4 target genes in SP21−HYP *vs.* LP21−HYP. (**B**) The network contained 10 down-regulated miRNAs and 8 target genes in SP21−HYP *vs.* LP21−HYP. Red and blue indicate up or down-regulation, respectively. (**C**) The network contained 14 upregulated miRNAs and target genes in SP21−PD *vs.* LP21−PD. (**D**) The network contained 6 downregulated miRNAs and target genes in SP21−PD *vs.* LP21−PD. Blue and Red indicate up or down-regulation, respectively.

**Figure 7 genes-13-01091-f007:**
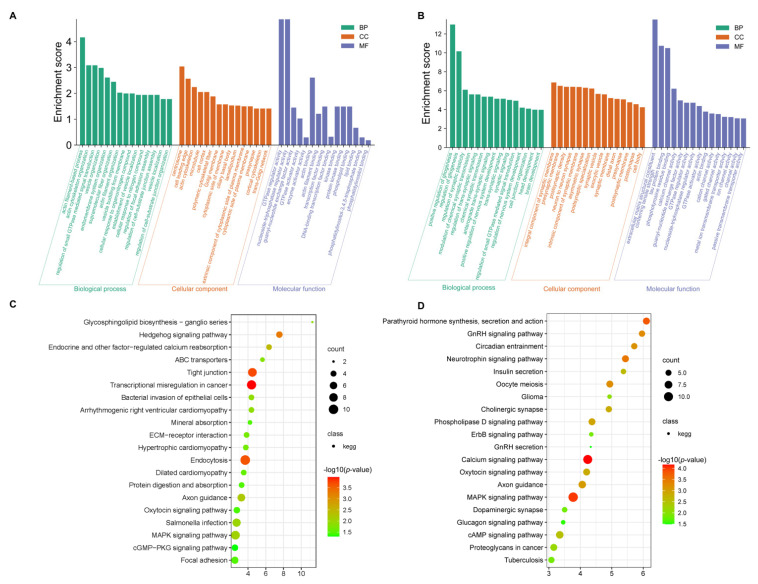
TOP enriched gene ontology (GO) study of target genes for differential expressed miRNAs between short-photoperiod and long-photoperiod in the hypothalamus and distal pituitary of sheep. (**A**) GO enrichment terms for target genes of differentially expressed miRNAs in SP21−HYP *vs.* LP21−HYP. (**B**) GO enrichment terms for target genes of differentially expressed miRNAs in SP21 −PD *vs.* LP21−PD. KEGG pathway analysis for target genes of differentially expressed miRNAs between short photoperiod and long photoperiod in hypothalamus and distal pituitary of sheep. (**C**) 20 enrichment pathways in SP21−HYP *vs.* LP21−HYP. (**D**) 20 enrichment pathways in SP21−PD *vs.* LP21−PD. Rich factors are defined as the amount of differentially expressed genes enriched in the pathway/amount of all genes in the background gene set.

**Figure 8 genes-13-01091-f008:**
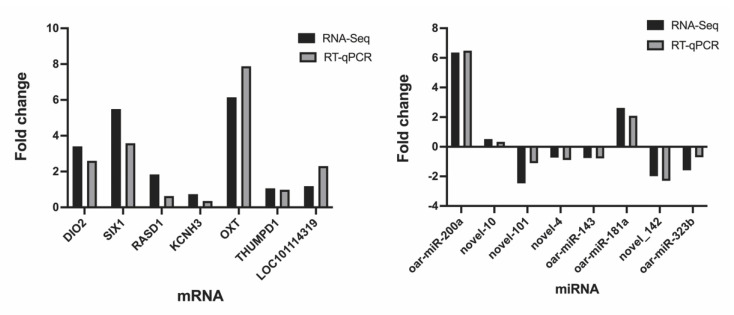
qPCR validation of mRNAs and miRNAs identified by RNA−seq.

**Table 1 genes-13-01091-t001:** The Details of Primers of mRNA.

Gene	Primer Sequences (5′-3′)	Product Size (bp)	Tm (℃)
*KCNH3*	F:CGCAGAACACCTTCCTGGACAC	122	60
	R:GCAGAAGCCATCAGAGCAGTAGAC		
*SIX1*	F:CATCGTTCGGCTTCACACAGGAG	142	60
	R:GCCTTGAGCACGCTTTCATTCTTG		
*DIO2*	F:CCAGAGCTGTTCCAAGGCAAGTC	109	60
	R:CTCCAGTGCTGCTGTCCAAGATG		
*RASD1*	F: GGAGACGTGTTCATCCTGGTGTTC	86	60
	R: GTGTCGAGAATCTGCCGCTTGAG		
*OXT*	F:GCCTTCTCCCAGCACTGAGA	81	60
	R: TCCTGGGGATGATTACAGAGGGA		
*THUMPD1*	F:TCAACGAATACGGCGACGACATG	108	60
	R:TCTTCAAGGCAGCTTCCACATCATC		
*LOC101114319*	F:GTACACCTTGGTCTTGACAGATCCG	121	60
	R: GAGAACCGTGCCACTGCTGATG		
*RPL19*	F:ATCGCCAATGCCAACTC	154	60
	R: CCTTTCGCTTACCTATACC		

**Table 2 genes-13-01091-t002:** The Details of Primers of miRNAs.

The Name of the Primer	Primer Sequence	Tm (℃)
oar-miR-200a	F:GCTGCAACACTGTCTGGTAACGAT	60
novel_10	F:CGCTTCACAGTGGCTAAGTTCTGC	60
novel_101	F:AGCTCTGGGTCTGTGGGGA	60
novel_4	F:CCGCGTCTTTGGTTATCTAGCTGTATG	60
oar-miR-143	F:CGCTGAGATGAAGCACTGTAGCTC	60
novel_142	F:CCACCTCCCCTGCAAACG	60
oar-miR-181a	F:TGCGAACATTCAACGCTGTCGGTGAG	60
oar-miR-323b	F:AGGCCTCCCAATACACGGTCGATCTC	60
U6	F:CAAGGATGACACGCAAATTCG	60

Note: The reverse primer for miRNA is the universal primer of the miRcute Plus miRNA qPCR Kit (TIANGEN, Beijing, China).

**Table 3 genes-13-01091-t003:** mRNAs with the Most Significant Differences in Expression of Hypothalamus and Distal Pituitary.

Group	mRNA	log2FoldChange	*q*-Value	Up/Down
SP21−HYP *vs.* LP21−HYP	*COMMD5*	−7.48	3.93 × 10^−72^	down
	*EHBP1*	−6.30	1.54 × 10^−^^51^	down
	*LOC114108752*	3.92	2.39 × 10^−^^48^	up
	*LOC114116052*	5.90	8.05 × 10^−^^45^	up
	*MAPK6*	−14.07	5.16 × 10^−^^24^	down
	*MAX*	1.93	2.30 × 10^−^^23^	up
	*ASPSCR1*	6.50	2.49 × 10^−^^23^	up
	*STK38*	5.08-	2.49 × 10^−^^23^	up
	*SV2A*	−9.16	3.40 × 10^−^^23^	down
	*RBM42*	13.67	3.76 × 10^−^^21^	up
SP21−PD *vs.* LP21−PD	*LOC114116052*	5.43	4.39 × 10^−^^80^	up
	*LOC114108752*	3.84	1.15 × 10^−^^43^	up
	*PRUNE2*	9.87	6.13 × 10^−^^38^	up
	*AKAP9*	15.95	9.79 × 10^−^^36^	up
	*GGNBP2*	6.59	1.94 × 10^−^^26^	up
	*BROX*	−13.02	2.59 × 10^−^^21^	down
	*RAPGEF6*	13.83	2.03 × 10^−^^18^	up
	*PTEN*	−4.09-	6.83 × 10^−^^17^	down
	*PLEC*	13.59	6.83 × 10^−^^17^	up
	*CEP112*	−8.20	1.14 × 10^−^^16^	down

**Table 4 genes-13-01091-t004:** miRNAs with the Most Significant Differences in Expression of Hypothalamus and Distal Pituitary.

Group	miRNA	log2FoldChange	*p*-Value	Up/Down
SP21−HYP *vs.* LP21−HYP	novel_154	7.34	1.17 × 10^−5^	up
	oar-miR-3956-5p	−2.32	1.33 × 10^−5^	down
	oar-miR-544-5p	3.86	3.61 × 10^−5^	up
	novel_156	4.64	6.50 × 10^−5^	up
	oar-miR-3956-3p	−3.68	1.61 × 10^−4^	down
	oar-miR-376e-3p	4.85	1.63 × 10^−4^	up
	oar-miR-376b-3p	5.55	1.70 × 10^−4^	up
	oar-miR-374a	3.85	1.77 × 10^−4^	up
	novel_234	3.86	3.66 × 10^−4^	up
	novel_199	5.67	6.00 × 10^−4^	up
SP21−PD *vs.* LP21−PD	novel_5	8.83	1.02 × 10^−9^	up
	novel_200	5.59	3.06 × 10^−9^	up
	novel_62	9.67	1.85 × 10^−6^	up
	novel_652	5.47	5.10 × 10^−6^	up
	novel_172	6.38	7.94 × 10^−6^	up
	novel_123	4.85	2.04 × 10^−5^	up
	novel_203	5.69	4.00 × 10^−5^	up
	novel_505	4.12	1.18 × 10^−4^	up
	novel_357	6.41	1.78 × 10^−4^	up
	novel_83	3.28	5.61 × 10^−4^	up

## Data Availability

Not applicable.

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
