# Peer review of "Screening of Differentially Expressed Genes and miRNAs in Hypothalamus and Pituitary Gland of Sheep under Different Photoperiods"

_genes, 2022, doi:10.3390/genes13061091_

Round 1

Reviewer 1 Report

Authors present work measuring differential expression of mRNAs and miRNAs in sheep hypothalamus and distal pituitary regions after animals were exposed to a shortened photoperiod or an extended photoperiod.  Authors hypothesized that they would find differential expression related to reproduction since the regions sampled are important to regulation of reproduction and photoperiod is an important external stimulus for reproduction.  Work completed to perform differential expression analyses, and in particular incorporating analyses of miRNAs, is interesting and valuable, but the paper did not fully convince me that its value derived from contributions to knowledge about reproduction.  Clarifications in primarily the discussion of results are necessary in reporting results of analyses to either fully highlight the scope of DE genomic elements or to focus solely on reproduction-related DE.

Major Comments:

More background information about how quickly light changes affect reproduction would be useful to put the work in context.  As the paper is currently written, it does not convince me that 21 days of photoperiod disruption is long enough to disrupt or change reproductive behaviors.

Table 4 miRNA labels should be more informative and information in the supplement should identify the novel miRNAs.  Useful information should at least include miRNA sequence and miRNA location in the genome.

Per the previous point, authors should plan to deposit their sequencing data in a data repository to allow reproducibility.

On page 3, the long and short photoperiods are stated as having the same amount of light/dark time (8h light, 16h dark).  Is that correct?

In the discussion of two specific seasonal pathways (paragraph end of page 16 to start of page 17), authors must include more information about these genes if they want to highlight them.  It appears from the text that only one of the genes had significant changes in expression between the two groups, and for none of the genes is it clear in the text to what degree they showed changes in expression.  I would strongly recommend creating a figure that includes statistics for each gene in these pathways of interest to support the in-text information.

The paper as it currently stands performed a genome-wide scan of DEGs and DEMs and attempted to show that those results disproportionately included reproduction-related genomic elements.  However, overall, the results do not seem to overwhelmingly indicate reproduction-related changes in association with changes in photoperiod.  It appears that a variety of biological functionalities change between the two groups of sheep, which is expected given the complex and diverse processes known to change with sleep changes and light exposure changes.  To reconcile the disparity between reported results and discussion or results, authors should do one of three things: 1) broaden their interpretation of results to include non-reproduction changes, 2) include more analyses to support the focus on only reproduction-related results, such as substantial enrichment of reproduction-specific (not only reproduction-related) pathways, or 3) focus reported results to only reproduction-related results and make it clear that they are only focusing on that subset.

In general, a more in-depth discussion of results would substantially strengthen this paper.  Discussion of results as it currently stands is a bit superficial.  Some more specific suggestions about strengthening the discussion are included below.

Authors should address any potential factors other than reproduction-related changes that could be causing the observed differential expression.  For example, changes in expression related to calcium processing seem like they should be connected to photoperiod changes because of differences in vitamin D availability, some expression changes may be due to stress due to photoperiod changes, etc.

Similar to the previous comment, authors should consider pleiotropic behaviors of genes when interpreting their results.  For example, IGF1 is reported as a significantly DE gene, and IGF1 is well known to have a wide range of functions.  Generally, there is little discussion of interpreting differential expression through the scope of pleiotropy.

Authors should more deeply discuss differences in results between the hypothalamic and distal pituitary results and whether they match expectations from previous work in the literature.

Minor/Technical Comments:

The figure includes grammatical errors throughout, but primarily in the introduction, that should be corrected prior to publication

Figure 4A and B text is so small and blurry that it is unreadable.

Figure 4C and D text is too small and blurry

Figure 7A and B text is so small and blurry that it is unreadable.

Figure 7C and D text is too small and blurry

Figure 6A and B, I would suggest changing the color scheme to something other than red/green to make the figure more colorblind friendly

Author Response

We appreciate your careful reading of our paper and the positive comments mentioned above.

Point 1:

More background information about how quickly light changes affect reproduction would be useful to put the work in context. As the paper is currently written, it does not convince me that 21 days of photoperiod disruption is long enough to disrupt or change reproductive behaviors.

Response 1:

We appreciate your comments on the study. Regarding the background on how quickly light changes affect reproduction, our laboratory has done a lot of work, which has been described in detail in the literature27 and 28. The present study builds on previous work on seasonally estrous Sunite sheep and uses the same approach with Small-tailed sheep.

Point 2:

Table 4 miRNA labels should be more informative and information in the supplement should identify the novel miRNAs. Useful information should at least include miRNA sequence and miRNA location in the genome.

Response 2:

Thank you very much for your question. We have added the mature sequences of the top 10 most significant miRNAs identified in Table S5 in the appendix.

Point 3:

On page 3, the long and short photoperiods are stated as having the same amount of light/dark time (8h light, 16h dark). Is that correct?

Response 3:

We are very sorry for our incorrect writing,we have made corrections in the text.

Point 4:

In the discussion of two specific seasonal pathways (paragraph end of page 16 to start of page 17), authors must include more information about these genes if they want to highlight them. It appears from the text that only one of the genes had significant changes in expression between the two groups, and for none of the genes is it clear in the text to what degree they showed changes in expression. I would strongly recommend creating a figure that includes statistics for each gene in these pathways of interest to support the in-text information.

Response 4:

Thank you very much for your suggestion regarding the statistics of the genes in the pathway of interest, we have attached Table S6 and Table S7 for presentation. Regarding the changes in these genes, we have also added Schedule Table S12 to support the text.

Point 5:

Authors should address any potential factors other than reproduction-related changes that could be causing the observed differential expression.  For example, changes in expression related to calcium processing seem like they should be connected to photoperiod changes because of differences in vitamin D availability, some expression changes may be due to stress due to photoperiod changes, etc.

Response 5:

Thanks to your suggestions, we have made changes in the text and added references.

Point 6:

Similar to the previous comment, authors should consider pleiotropic behaviors of genes when interpreting their results. For example, IGF1 is reported as a significantly DE gene, and IGF1 is well known to have a wide range of functions. Generally, there is little discussion of interpreting differential expression through the scope of pleiotropy.

Response 6:

Thanks to your suggestions, the discussion of the gene IGF1 was removed.

Point 7:

Authors should more deeply discuss differences in results between the hypothalamic and distal pituitary results and whether they match expectations from previous work in the literature.

Response 7:

 Thanks to your suggestions, the discussion of the differences between hypothalamic and pituitary results is discussed in depth in the last paragraph, and we have added the supporting text of Schedule Table S12.

Point 8:

Minor/Technical Comments:

The figure includes grammatical errors throughout, but primarily in the introduction, that should be corrected prior to publication

Figure 4A and B text is so small and blurry that it is unreadable.

Figure 4C and D text is too small and blurry

Figure 7A and B text is so small and blurry that it is unreadable.

Figure 7C and D text is too small and blurry

Figure 6A and B, I would suggest changing the color scheme to something other than red/green to make the figure more colorblind friendly

Response 8:

Thanks to the reviewer for helpful comments, we have made changes to the clarity of the images in the revised version of the text to show the reader better, and we were careless with Figure 6A and B. We have redrawn it to be more color blind friendly.

Other changes:

  1. Lines 18, 26, and 31 of the text 215 and 216 were revised.
  2. The figure notes of Figure 2 were modified.
  3. We are very sorry for the data error in the Table 4 section, and we have corrected it.
  4. Figure 5, we replaced the font due to unattractive reasons.

Reviewer 2 Report

Very good and interesting data, congrats 

Author Response

Thank you for your careful reading of our paper and the positive comments above.

Reviewer 3 Report

Dear Authors,

The paper reviewed was prestented as a very high quality draft, only with small editorial errors, like RLP19 gene or PRL19 gene names etc. 

Best wishes,

Reviwer

Author Response

We appreciate your careful reading of our paper and the positive comments mentioned above. where the RPL19 gene has been corrected in the table.